# Extracellular Succinate: A Physiological Messenger and a Pathological Trigger

**DOI:** 10.3390/ijms241311165

**Published:** 2023-07-06

**Authors:** Kenneth K. Wu

**Affiliations:** 1Institute of Cellular and System Medicine, National Health Research Institutes, 35 Keyan Road, Zhunan, Miaoli County 35053, Taiwan; kkgo@nhri.org.tw; Tel.: +886-37-246-166; Fax: +886-37-587-408; 2Institute of Biotechnology, College of Life Science, National Tsing-Hua University, Hsinchu 30013, Taiwan; 3Graduate Institute of Basic Medical Science, China Medical University, Taichung 40402, Taiwan

**Keywords:** extracellular succinate, succinate receptor-1, succinate dehydrogenase, inflammation, fibrosis, myocardial infarction

## Abstract

When tissues are under physiological stresses, such as vigorous exercise and cold exposure, skeletal muscle cells secrete succinate into the extracellular space for adaptation and survival. By contrast, environmental toxins and injurious agents induce cellular secretion of succinate to damage tissues, trigger inflammation, and induce tissue fibrosis. Extracellular succinate induces cellular changes and tissue adaptation or damage by ligating cell surface succinate receptor-1 (SUCNR-1) and activating downstream signaling pathways and transcriptional programs. Since SUCNR-1 mediates not only pathological processes but also physiological functions, targeting it for drug development is hampered by incomplete knowledge about the characteristics of its physiological vs. pathological actions. This review summarizes the current status of extracellular succinate in health and disease and discusses the underlying mechanisms and therapeutic implications.

## 1. Introduction

Succinic acid is a tricarboxylic acid (TCA) cycle metabolite that is derived from succinyl CoA and oxidized to form fumarate [1]. It is normally confined to the mitochondrial matrix and its level is tightly regulated [1]. When cells are stressed, disruption of the TCA cycle may result in elevation of succinate in the matrix and the leakage of excessive succinate into the cytoplasm, where it acts as a signaling molecule to impact diverse cellular functions through inhibition of a large group of 2-oxoglutarate-dependent dioxygenases (2OGDD), notably prolyl hydroxylase (PHD) and ten–eleven translocation (TET) [2,3]. Inhibition of PHD by cytosolic succinate leads to impaired degradation of hypoxia inducible factor-1 (HIF-1α) and HIF-1α-mediated changes in transcription of metabolic enzymes, angiogenic factors, and pro-inflammatory mediators [4,5,6,7]. Inhibition of TET-2, on the other hand, results in impaired hydroxylation of DNA methyl groups, and, consequently, DNA hypermethylation, which is associated with tumorigenesis and cancer metastasis [8,9]. Cytosolic succinate is secreted into the extracellular space and diffused into the circulating blood, where it acts as a local and/or systemic autacoid, regulating physiological functions and pathological processes. Cytosolic succinate has been covered in excellent review articles [10,11]. This paper will focus on extracellular circulating succinate with respect to its roles as physiological messengers and pathological triggers, its mechanisms of action, and its potential as a target for new therapeutic strategies.

## 2. Skeletal Muscle-Derived Extracellular Succinate Confers Physiological Adaptation to Exercise and Cold Exposure

Extracellular succinate is a crucial messenger to facilitate adaptation to physiological stresses, such as cold exposure, vigorous exercise, and physical activity. Skeletal muscle plays a central role in providing extracellular succinate to drive the adaptation.

### 2.1. Extracellular Succinate Promotes Muscle Remodeling

It was reported almost a half century ago that aerobic exercise increased blood levels of succinate [12]. Subsequent studies confirmed the accumulation of succinate during exercise. A meta-analysis of relevant exercise-associated metabolomic studies reveals that exercise increase circulating metabolites, including succinate [13]. A recent report indicates that, in response to exercise, succinate is secreted from skeletal muscle cells via monocarboxylate transporter-1 (MCT-1), and the secreted succinate activates non-myofibrillar resident cells such as stromal cells, endothelial cells, and satellite cells in skeletal muscle to promote skeletal muscle remodeling and innervation via succinate receptor-1 (SUCNR-1) [14]. It is unclear how exercise induces succinate accumulation. One possible explanation is that exercise induces muscle cell metabolic reprogramming, leading to increased glycolysis and a disrupted TCA cycle, such as inhibition of succinate dehydrogenase (SDH). SDH possesses dual activities: it catalyzes oxidation of succinate to form fumarate in the TCA cycle and functions as complex II in the ETC to convert ubiquinone to ubiquinol for oxidative phosphorylation and ATP generation [15,16]. Akin to exercise-induced skeletal muscle metabolic changes, LPS-stimulated macrophages exhibit a metabolic shift to aerobic glycolysis and perturbation of the TCA cycle characterized by blocking isocitrate dehydrogenase activity, which leads to itaconate accumulation [17]. Excessive itaconate inhibits SDH catalytic activity, resulting in succinate accumulation [17]. Contrary to this possible mechanism is a report that congenital deficiency of SDH in a patient rendered intolerance to exercise due to early skeletal muscle fatigue and weakness during exercise [18,19]. It was proposed that TCA enzyme deficiency impairs oxidative phosphorylation and reduces energy supply. Succinate supplementation was reported to enhance muscle fiber oxidative phosphorylation and increase muscle strength and endurance [20,21]. The reason for the different roles that extracellular succinate plays in muscle strength and endurance during exercise is unclear. Further studies are needed to elucidate the underlying mechanism.

### 2.2. Extracellular Succinate Upregulates Adipose Tissue Energy Expenditure and Thermogenesis

Brown adipose tissues (BATs) act as a central regulator of energy expenditure for thermogenesis and play an important role in maintaining body temperature during cold exposure [22,23]. Metabolomic analysis identifies succinate as a key driver of energy expenditure in BAT [24]. It has been reported that cold-exposure-associated muscular shivering leads to release of succinate from muscle fibers, resulting in elevation of succinate levels in the circulating blood [24]. The extracellular succinate is taken up by brown adipocytes and enters the mitochondria, where it replenishes the substrate for SDH and complex II to generate ROS and stimulate uncoupled respiration in an uncoupled protein-1 (UCP-1)-dependent manner [24]. Thermogenesis from the UCP-1 pathway is abrogated by either pharmacological inhibition of muscle contraction or SDH/complex II. It is to be noted that BAT UCP-1 is a key player in controlling the level of succinate in circulating blood. Genetic deletion of UCP-1 in mice results in the elevation of blood succinate levels, which contributes to liver inflammation and fibrosis [25].

In summary, skeletal muscle cells are in a pivotal position to supply extracellular succinate, which acts as an intercellular messenger to enhance muscle endurance during vigorous exercise and generate heat during cold exposure (Figure 1).

## 3. Extracellular Succinate Mediates Diverse Pathophysiological Processes via SUCNR-1

As described above, extracellular succinate serves as a substrate supplement via anaplerosis to confer thermogenesis for adaptation to cold and muscle remodeling to enhance endurance during vigorous exercise. However, extracellular succinate mediates diverse pathophysiological processes and exacerbates human diseases through interaction with a membrane G-protein-coupled receptor (GPCR), GPR91. In searching for natural ligands for orphan GPCRs, He et al. identified succinate from animal kidney extracts as a selective ligand for GPR91, an orphan GPCR with sequence homology to purinergic receptors [26,27,28]. GPR91 was shown to be expressed in the juxtaglomerular apparatus, especially the afferent arterial endothelial cells, and involved in inducing renin production and secretion [29,30]. It was linked to diabetes-associated hypertension, as elevated succinate in diabetes stimulates renin production via GPR91 [29,30]. GPR91 was detected in proximal and distal duct epithelial cells in kidneys and was considered to play a role in regulating renal tubular functions, although the exact activity remains to be ascertained [31]. Further studies reveal that GPR91 expression is not restricted to renal tissues. In fact, it is widely distributed on different cells where it carries out tissue-specific functions and mediates pathophysiological processes in human diseases. As GPR91 proves to be a selective succinate receptor, it is commonly called SUCNR-1. Succinate-ligated SUCNR-1 is signaled via Gi and Gq with the activation of cardinal signaling pathways, resulting in elevation of intracellular calcium and inositol triphosphate (IP_3_), reduction of cyclic AMP, and activation of ERK1/2 [27]. Under certain circumstances, succinate-activated SUCNR-1 may signal via Gq, as reported in macrophages by [32]. SUCNR-1 mediates physiological roles, such as immune responses, glucose homeostasis, hematopoiesis, and platelet aggregation, and pathological conditions, such as tissue injury, inflammation, fibrosis, and cancer metastasis [33,34,35,36,37,38]. The succinate-SUCNR-1 axis plays a critical role in important human diseases (Figure 2).

## 4. Succinate Elicits Intestinal Immunity and Triggers Colon Inflammation

Under normal conditions, the intestinal lumen and feces contain low levels of succinate, despite a large quantity of succinate produced by microbial fermentation of dietary fibers [39]. Low luminal succinate is attributed to the conversion of succinate to short-chain fatty acids (acetate, propionate, and butyrate) by succinate-consuming bacteria in the gut microbiota [40,41]. Intestinal epithelial cells express solute carrier 13A (SLC13A) family proteins, notably SLC13A-2, -3, and -5, which take up luminal succinate [42]. It was reported that small intestinal epithelial cells take up succinate and convert it to glucose via gluconeogenesis [43], which exerts a great impact on reducing lipid deposition and attenuating hepatic inflammation [44]. Despite a low level of succinate in the lumen of the small intestine, succinate derived from the diet or protist and helminth infestation interacts with SUCNR-1 expressed on the tuft cells and transmits signals to secrete IL-25, which induces tuft cell and goblet cell hypertrophy and activates type 2 innate lymphoid cells (ILC2) to elicit type II immunity [45,46,47].

By contrast, succinate accumulation in the colon exerts a detrimental effect on the epithelium. Perturbation of colon microbiota by antibiotics, high-fat diet, or inflammatory mediators results in increasing succinate-producing and diminishing succinate-consuming strains in the colon microbiome and the consequent overproduction of succinate [48,49]. Unlike the small intestine, which contains succinate-sensing tuft cells, the colon epithelium contains few, if any, tuft cells and does not elicit immunity. The accumulated luminal succinate disrupts epithelial barrier function and induces inflammation and fibrosis by activating macrophages and fibroblasts, which reside in the subepithelial region [49]. Chronic succinate elevation in the colon lumen was reported to induce colitis in mice [49]. Succinate causes epithelial damage, macrophage activation, and fibroblast transdifferentiation by ligating SUCNR-1 and activating SUCNR-1-mediated signaling pathways. One of the mechanisms by which luminal succinate reaches the subepithelial space is through a transepithelial pathway in which succinate is taken up by epithelial cells via Slc13A transporters, such as Slc13A-2 or A-3, and secreted into the extracellular space via the organic ion transporters [42]. Extracellular succinate is taken up by the infiltrating macrophages, and elevated cytosolic succinate enhances inflammation through inhibition of PHD, thereby stabilizing HIF-1α, which mediates IL-1β release [7,50]. As illustrated in Figure 3, microbiota-derived succinate is handled differently in the small intestine vs. the colon, carries out physiological functions in the small intestine via tuft cells, and induces colon epithelial cell damage and subepithelial inflammation and fibrosis by activating macrophages and fibroblasts.

Serum and fecal succinate levels are elevated in humans with inflammatory bowel diseases, i.e., Crohn’s disease and ulcerative colitis [42,49]. SUCNR-1 is detected on epithelial cells as well as inflammatory cells and fibroblasts in the lamina propria of IBD colonic tissues [42,49]. Importantly, SUCNR-1 expression is correlated with fibrosis [49]. Taken together, microbiota-derived succinate plays an important role in aggravating inflammation and inducing fibrosis in human IBD via SUCNR-1.

## 5. Ischemia-Reperfusion (I/R)-Derived Succinate Contributes to Myocardial Infarction (MI)

Experimental results have shown that I/R injury to vital organs may result in succinate accumulation and secretion. Comparative metabolomic analysis of I/R-injured cardiomyocytes has provided valuable information regarding the source of succinate accumulation [51]. I/R injury was reported to cause perturbation of mitochondrial metabolism with overflow of fumarate generated from the malate–aspartate shunt and the purine nucleotide cycle [51]. Fumarate is reduced to succinate by the reversal of SDH activity [51]. It was subsequently reported that cardiac ischemia augments the canonical TCA cycle, which may be the major source of succinate accumulation [52]. It is likely that succinate accumulation is attributable to both sources. I/R injury disturbs the mitochondrial ETC and causes reversed electron transport at complex I of the ETC, resulting in ROS accumulation [51].

Accumulated succinate in the mitochondrial matrix of cardiomyocytes is released into the cytoplasm and secreted into the extracellular space via MCT-1 [53], which accounts for the elevation of circulating succinate levels in patients with acute MI [54]. Elevated circulating succinate is correlated with the extent of myocardial injury and is considered a biomarker of I/R injury. I/R-induced succinate secretion was reported in other tissues and cells, including hepatocytes and neurons [55,56]. The extracellular succinate acts in an autocrine or paracrine manner to damage cells. It was reported that succinate mediates ischemia-induced cardiomyocyte apoptosis by altering mitochondrial dynamics; it triggers Drp-1-dependent mitochondrial fission and induces mitochondrial fragmentation, which results in mitochondrial dysfunction and apoptosis [57]. Succinate elicits mitochondrial dysfunction and apoptosis through interaction with SUCNR-1 on the cardiomyocyte surface, which signals via ERK1/2 [57].

Post-MI heart failure has emerged as a serious human disease. Post-MI cardiac damage is characterized by persistent cardiomyocyte death, myocardiocyte hypertrophy, myocardial inflammation, and fibrosis, which leads to structural remodeling and functional failure. Extracellular succinate was considered to play an important role in post-MI heart failure. It has been reported that succinate induces myocardiocyte hypertrophy via SUCNR-1 [58]. As will be discussed in the next sections, extracellular succinate contributes to macrophage recruitment and activation and is a key metabolite in enhancing inflammatory responses in injured tissues. Furthermore, succinate has been shown to attenuate post-injury tissue fibrosis by reducing fibroblast differentiation to myofibroblasts, either directly or indirectly, via controlling macrophage-mediated fibroblast activation and differentiation. Thus, extracellular succinate is a key player in I/R-induced myocardial infarction and post-MI heart failure through its actions on apoptosis, inflammation, and fibrosis.

## 6. Extracellular Succinate Aggravates Inflammatory Responses through Macrophage Activation

Succinate released into the tissue damage site or pro-inflammatory microenvironment acts as a proinflammatory factor to attract macrophages, enhance macrophage migration, and induce macrophage activation. It has been reported that succinate activates and induces M1 macrophages to release pro-inflammatory cytokines and chemokines by ligating membrane SUCNR-1 and activating the downstream signaling pathways and transcriptional program [10]. Of note, macrophages are the target of succinate and also the source of succinate secretion. For example, LPS-stimulated macrophages undergo metabolic reprogramming, which leads to succinate accumulation and secretion [17]. The extracellular succinate-SUCNR-1 pathway contributes to diverse inflammatory disorders and aggravates their inflammatory manifestations. By contrast, extracellular succinate may confer anti-inflammatory messages. Highlighted below are the pro-inflammatory and anti-inflammatory properties.

### 6.1. Activated M1 Macrophages Secrete Succinate to Enhance Inflammatory Response and Exacerbate Inflammatory Disorders

Interferon-γ and LPS convert naïve macrophages into an M1 phenotype [59,60]. Metabolic analysis of LPS-induced M1 macrophages reveals a metabolic shift from oxidative phosphorylation to aerobic glycolysis, which is accompanied by a break in the mitochondrial TCA cycle at isocitrate dehydrogenase (IDH) [17]. IDH catalyzes the conversion of isocitrate to α-ketoglutarate (α-KG), and its breakdown results in the accumulation of isocitrate, which is further converted to itaconate. Itaconate was reported to inhibit SDH, resulting in the accumulation of succinate [61,62].

In inflammatory disorders, such as rheumatoid arthritis, macrophage-released succinate upregulates SUCNR-1 expression on M1 macrophages and stimulates IL-1β production via SUCNR-1 to exacerbate arthritis [63]. Succinate accumulation in LPS-treated macrophages induces IL-1β expression through HIF-1α [7]. LPS-induced M1 macrophages and microglial cells exhibit changes in mitochondrial dynamics, shifting toward mitochondrial fission and fragmentation [64,65]. Mitochondrial fragmentation is pivotal in pro-inflammatory cytokine production [64]. In addition, it enhances succinate secretion [65]. Drp-1-dependent mitochondrial fragmentation and succinate accumulation may form a regulatory loop to enhance the activity of pro-inflammatory macrophages. This notion requires proof through additional experiments.

### 6.2. Adipose Tissues Release Succinate to Recruit Macrophages and Aggravate Inflammation in Obesity

Adipose tissue inflammation is a cardinal manifestation of obesity that contributes to insulin resistance and diabetes mellitus. Obese adipocytes were reported to release succinate into the extracellular space and increase the succinate level in circulating blood [66]. Gene expression profiling reveals that succinate enhances inflammatory responses via a SUCNR-1-mediated signaling pathway [66]. Succinate increases macrophage infiltration in obese adipose tissues through chemotaxis and promotion of macrophage migration [66]. The effect of succinate on macrophage chemotaxis, migration, and infiltration depends on SUCNR-1, as macrophage infiltration and adipose tissue inflammation are considerably reduced in SUCNR-1 knockout mice [66].

### 6.3. Succinate-SUCNR-1 Axis Is Involved in Macrophage M2 Polarization and Anti-Inflammatory Actions

Although the succinate-SUCNR-1 axis plays a crucial role in mediating inflammatory responses and aggravating inflammatory disorders, there are reports indicating that this axis is critical in driving macrophage M2 polarization and inflammation control. A recent report from our group indicates that succinate induces macrophage M2 polarization [67]. Furthermore, cancer cell-derived succinate polarizes macrophages into M2-like tumor-associated macrophages, which contribute to cancer cell migration and cancer metastasis [67]. Succinate induces M2 polarization by SUCNR-1-mediated signaling via the PI-3K/Akt pathway [67]. In addition to inducting M2 polarization, extracellular succinate hyperpolarizes M2 macrophages by binding to SUCNR-1, expressed abundantly on M2 macrophages, and signaling via the Gq pathway [32]. The succinate-SUCNR-1-M2 polarization pathway was reported to control inflammation in adipose tissues in healthy, lean human subjects [68]. In murine models, myeloid-specific deletion of SUCNR-1 promotes inflammation and alters adipose tissue fat cell distribution, with a reduction of brown adipocytes [68].

### 6.4. Succinate Controls Inflammatory Response through Cell–Cell Interaction

Extracellular succinate may act as an intercellular messenger to regulate inflammation. It was reported that macrophage-derived succinate interacts with transplanted neural stem cells (NSC) to suppress inflammation in a multiple sclerosis (MS) murine model [69]. Macrophages infiltrated in MS tissues secrete succinate, which interacts with SUCNR-1 expressed on NSCs. This leads to transactivation of *ptgs2* coding for cyclooxygenase-2 (COX-2, also known as prostaglandin H synthase-2, PGHS-2). COX-2 and prostaglandin E synthase catalyze the synthesis of PGE_2_, which is released into the extracellular space, where it suppresses macrophage activation and pro-inflammatory cytokine production via PGE_2_-specific receptors [70,71]. In addition, macrophage-derived succinate interacts with NSC SUCNR-1 to activate SLC13 transcription and enhance SLC13 expression, which takes up succinate and reduces the level of extracellular succinate. Transplantation of NSCs into the MS model was previously reported to be effective in controlling cerebral inflammation, but the underlying mechanism was unclear [72,73]. Findings from this report suggest that the anti-inflammatory effect of NSCs is attributable to macrophage-derived extracellular succinate, which acts as an inflammation suppressor.

## 7. Injured Hepatocytes Secrete Succinate to Activate Hepatic Stellate Cells (HSC) and Induce Fibrosis

Liver fibrosis signifies an advanced stage of liver diseases, including liver cirrhosis and non-alcoholic steatohepatitis (NASH), and is often associated with liver failure [74,75,76,77,78,79]. Liver fibrosis is orchestrated by macrophages and HSCs. A selective subset of macrophages secretes pro-fibrotic factors to transdifferentiate HSCs [75]. Extracellular succinate is identified as one of the important pro-fibrotic factors. Liver tissues injured by ischemia release succinate, which activates adjacent HSCs [56]. HSCs are located in the subendothelial space and contact the basolateral surface of hepatocytes. Succinate released from injured hepatocytes enhances adjacent HSC migration and proliferation [80] and induces HSC activation and differentiation by interacting with HSC surface SUCNR-1 [56], which signals the expression of α-SMA and collagen I in HSC, thereby increasing extracellular matrix deposition [81,82]. SUCNR-1 expression is upregulated in activated HSCs [79]. Suppression of SUCNR-1 expression by siRNA abrogates α-SMA expression. The succinate–SUCNR-1 axis plays a pivotal role in liver fibrosis following hepatic damage.

It is important to note that extracellular succinate suppresses SDH expression, thereby augmenting succinate accumulation and secretion [81,82]. Inhibition of SDH with malonate increases succinate accumulation, accompanied by enhanced expression of α-SMA. HSCs cultured in a medium simulating a NASH-producing diet, such as a methionine- and choline-free medium or high-palmitate medium, exhibit reduced SDH activity and an increased succinate level, accompanied by elevated myofibroblast markers [83]. Mice given a NASH diet have reduced SDH expression and increased succinate levels. It was reported that succinate suppresses SDH activity by reducing sirtuin 3 (Sirt3) [84]. Sirt3 binds to the SDHA subunit and enhances SDH catalytic activity through control of lysine acetylation [84,85]. In HSC cellular experiments and NASH animal models, succinate was reported to reduce Sirt3, accompanied by suppression of SDH and an increase in SUNCR-1 and succinate [84]. Furthermore, Sirt3 reduction is correlated with SDH suppression. Taken together, the experimental findings suggest that SDH suppression, extracellular succinate accumulation, and SUCNR-1 upregulation form a vicious cycle to enhance HSC activation and myofibroblast transdifferentiation and perpetuate myofibroblast-mediated liver fibrosis following hepatic damage by viral infection, toxin exposure, and/or a high-fat diet (Figure 4). This vicious cycle is likely to drive tissue damage and fibrosis in other vital organs subjected to I/R and toxic injuries.

## 8. Cancer Cell-Derived Succinate Acts as a Messenger in Tumor Microenvironment to Educate Stromal Cells and Promote Cancer Progression

Wu et al. from our group reported that lung cancer cells secrete succinate, which induces M2-like TAMs and promotes cancer cell migration and metastasis [67]. Other types of cancer cells, such as gastric cancer cells, were reported to secrete succinate to increase angiogenesis and promote cancer growth [86]. The mechanism by which succinate is accumulated and secreted into the microenvironment has not been fully elucidated, but several reports imply that reduced expression of SDH subunits and the consequent loss of SDH activity may account for succinate accumulation. For example, reduced expression of the SDHB or SDHD subunit was reported in colorectal, gastric, hepatocellular, ovarian, and clear-cell renal cell carcinoma [8,85,86,87]. Silencing of cancer cell SDHB was reported to result in loss of SDH activity and a change in cancer cell behavior; cells are more migratory and exhibit epithelial mesenchymal transition (EMT) [87,88,89,90]. Of note, mutations of SDH subunits, notably SDHB and SDHD, are detected in hereditary paragangliomas and pheochromocytomas [91,92]. Subunit B mutation results in loss of SDH catalytic activity and an increase in angiogenesis via cytosolic succinate accumulation, which increases HIF-1α [93]. Tumors bearing SDHB mutations are prone to malignant transformation and metastasis [94,95]. Paraganglioma and pheochromocytoma cells express SUCNR-1. Genetic deletion of SUCNR-1 results in reduced tumor growth and metastasis, suggesting that tumor-released succinate promotes tumor growth through activation of SUCNR-1 [94,96,97]. Taken together, the reported data suggest that SDHB mutation or expression defect is a major source of succinate accumulation and secretion. It should be mentioned that TRAP-1 plays a role in controlling SDH activity. TRAP-1 expression is often increased in cancer cells, which has been shown to inhibit SDH activity [98,99,100].

Succinate in the tumor microenvironment (TME) acts in a paracrine manner to enhance cancer cell migration and induce EMT through interaction with SUCNR-1 and the downstream PI-3K/Akt signaling pathway [67]. It activates endothelial cells and promotes angiogenesis [84]. Furthermore, as described above, it polarizes macrophages into M2-like tumor-associated macrophages (TAM) [32]. It may also affect other stromal cells in TME, such as fibroblasts and T-lymphocytes. It is to be noted that macrophages may secrete succinate into TME to increase TME succinate concentrations. Succinate-induced HIF elevation contributes to cancer metastasis by upregulating TWIST expression, triggering SNAIL translocation to the nucleus [101,102,103,104,105], and promoting vascular endothelial growth factor (VEGF) expression, which enhances cancer cell migration, EMT, and metastasis [106,107]. Thus, TME succinate is one of the key messengers to promote cancer growth and, particularly, cancer metastasis (Figure 5).

## 9. Succinate-SUCNR1 Axis as a Therapeutic Target

Based on experimental findings from animal and cellular studies, a number of succinate-related therapeutic targets have been proposed. They include targeting SUCNR-1 or SDH or infusion of succinate, among which, targeting the succinate-SUCNR-1 axis has been more extensively investigated. Although experimental work suggests that inhibition of SDH with dimethylmalonate protects against I/R-induced cardiac damage, ROS accumulation, and inflammation [108,109,110], it is unclear how SDH inhibitors exert anti-inflammatory and tissue protection effects. Succinate was proposed to have a beneficial effect against severe sepsis, primarily through supplementation of succinate for ATP generation, which rescues tissues from sepsis-associated bioenergetic facture [111,112,113,114]. However, logistical translation of the experimental animal studies into clinical therapeutic use is confronted by major hurdles [115,116].

As highlighted above, the succinate-SUCNR-1 axis is implicated in mediating or aggravating diverse pathological conditions and diseases, including inflammatory disorders, I/R injury-mediated organ infarction and post-injury structural remodeling and functional failure, and cancer progression and metastasis. SUCNR-1 is considered a viable therapeutic target [117]. As the knowledge of the SUCNR-1 structure–function relationship and pathophysiological roles is expanding, structure-based novel antagonists have been developed [118,119]. These novel compounds have the potential to be therapeutic agents against inflammation-related diseases and cancer metastasis. However, it should be noted that SUCNR-1 mediates important physiological functions and represents a key metabolite in maintaining energy homeostasis; systemic use of SUCNR-1 antagonists may be confronted by severe adverse effects. Furthermore, it has been reported that SUCNR-1 activation confers macrophage anti-inflammatory properties, while SUCNR-1 deficiency promotes inflammation [68]. How to use the SUCNR-1 antagonists judiciously in treating SUCNR-1-mediated diseases without jeopardizing normal or beneficial SUCNR-1 actions is not well understood. Further studies are needed to sort out the different characteristics between “good” and “evil” SUCNR-1.

## 10. Blood Levels of Succinate in Health and Disease

As highlighted in the previous sections, extracellular succinate acts in an autocrine and/or paracrine fashion to carry out physiological and pathological functions. Importantly, it is a circulating hormone. Normal human blood contains micromolar concentrations of succinate. The reported blood concentrations of succinate from several studies varied with the mean concentrations, ranging from 6.1 μM to 23.5 μM [120]. The variations are not surprising, as the blood succinate concentration of healthy subjects is regulated by succinate production and influenced by diverse environmental factors, including diet, physical activity, and cold temperature [13,24,48,121]. Succinate production by several organs, including the intestine, heart, muscle, and liver, contributes to the circulating succinate in healthy subjects [14,54,122]. Under stressful and/or pathological conditions, such as I/R injury and endotoxemia, succinate production is increased, which leads to the elevation of circulating succinate levels [54,123]. Of note, a high-fat diet, antibiotic abuse, and intestinal inflammation alter the composition and metabolism of intestinal microbiota, resulting in overproduction of intestinal succinate and elevation of circulating succinate levels, which may cause local and systemic inflammation [48]. A high-fat and high-sucrose diet was reported to induce obesity and elevate circulating succinate levels in experimental animals [121]. In fact, elevated circulating succinate has been reported in several disease animal models, including obesity, hypertension, and metabolic disease [124]. Circulating succinate plays important physiological roles. However, elevation of blood succinate levels is associated with severe human diseases, such as sepsis and cardiovascular disease, and is considered a risk factor and biomarker [123,125]. It should be mentioned that our understanding of circulating succinate remains incomplete. Further studies are needed to characterize its regulatory mechanisms and systemic actions and to ascertain its clinical use as a diagnostic and therapeutic biomarker.

## 11. Conclusions

Cells under diverse stresses, ranging from vigorous exercise and cold exposure to bacterial infection and I/R injury, secrete succinate into the extracellular space and raise the circulating levels of succinate to assist tissue adaptation to the stressful conditions for survival. Under extreme stresses, such as endotoxemia-induced septic syndrome, succinate supplementation may rescue cells and tissues from energy depletion. However, extracellular succinate may turn from Jekyll to Hyde to harm the cells and damage tissues, leading to cell death and functional failure when tissues are subjected to persistent stresses, such as bacterial infection and ischemia-reperfusion injury. Although the reason for the switch from a physiological messenger to a tissue killer has not been fully elucidated, several factors, such as the nature of the injurious agents, the extent and duration of injury, and the microenvironment, contribute to the damaging effects. The Jekyll and Hyde roles of succinate hamper development of succinate-targeted therapy for debilitating human diseases, including sepsis, heart failure due to myocardial infarction, and a variety of inflammatory disorders. Further studies are needed to characterize the physiological vs. pathological roles of extracellular succinate and target the distinct modes of succinate action in causing cell death, inflammation, and fibrosis.

The actions of extracellular succinate are distinct from those of cytosolic succinate. A key distinction is that extracellular succinate acts via a specific membrane receptor, SUCNR-1, which signals via Gi-protein and Gq-protein-mediated signaling pathways [27]. In certain cell types, such as M2 macrophages, the action of extracellular succinate is signaled via Gq [32]. As SUCNR-1 is widely expressed in human cells, extracellular succinate induces a variety of pathological processes in different tissues under diverse stresses. Extracellular succinate perturbs fundamental cellular metabolism and functions. These fundamental changes account for various pathological lesions. Of note, both extracellular succinate and cytosolic succinate increase HIF-1α, albeit via different mechanisms. Extracellular succinate increases HIF-1α expression via the SUCNR-1→PI-3K/Akt signaling pathway. HIF-1α is a pleiotropic transcription factor that promotes the expression of genes for angiogenesis, glycolysis, cancer cell invasion and metastasis, and inflammation. Extracellular succinate acts in concert with cytosolic succinate to alter cellular metabolism and function and carry out physiological and pathological roles.

The pro-inflammatory and pro-fibrotic actions of extracellular succinate may be linked to mitochondrial dynamic changes. It is interesting to note that LPS-stimulated macrophages exhibit excessive mitochondrial fission accompanied by TCA cycle breaks at IDH and SDH steps, resulting in succinate, citrate, and itaconate accumulation. Succinate, in turn, induces mitochondrial fission and fragmentation, thus forming a vicious cycle. The regulatory loop of mitochondrial fragmentation via Drp-1 and succinate accumulation and secretion may be a major force in the damaged tissue microenvironment to drive inflammation and fibrosis. Mitochondrial fragmentation induced by cellular injury plays an important role in cardiac and renal dysfunction [126,127], and inhibition of mitochondrial fission has been shown to alleviate post-infarct cardiac structural damage and functional impairment [127,128]. Injury-induced mitochondrial fragmentation and the consequent fragmentation–succinate regulatory loop are targets for developing new therapeutic agents against inflammation, fibrosis, and organ failure. 

## Figures and Tables

**Figure 1 ijms-24-11165-f001:**
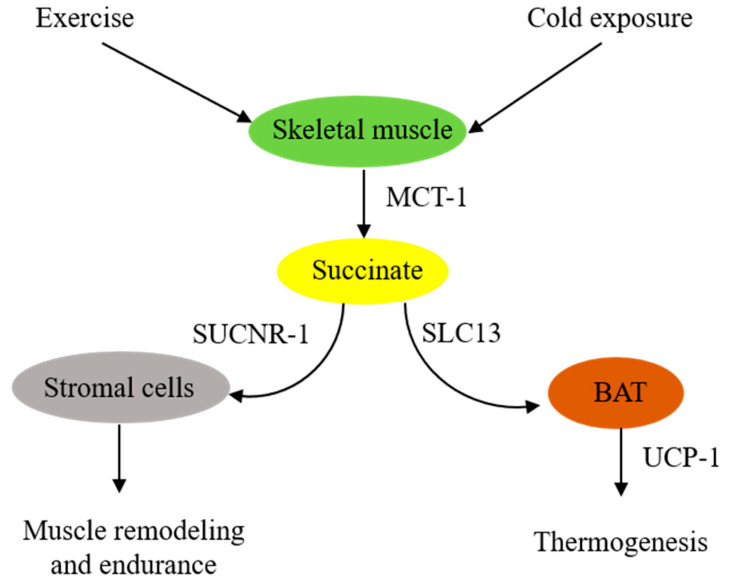
**Schematic illustration of extracellular succinate as a physiological messenger.** Physiological stresses such as vigorous exercise or cold exposure induce skeletal muscle fibers to secrete succinate, which acts as a messenger to confer physiological adaptation.

**Figure 2 ijms-24-11165-f002:**
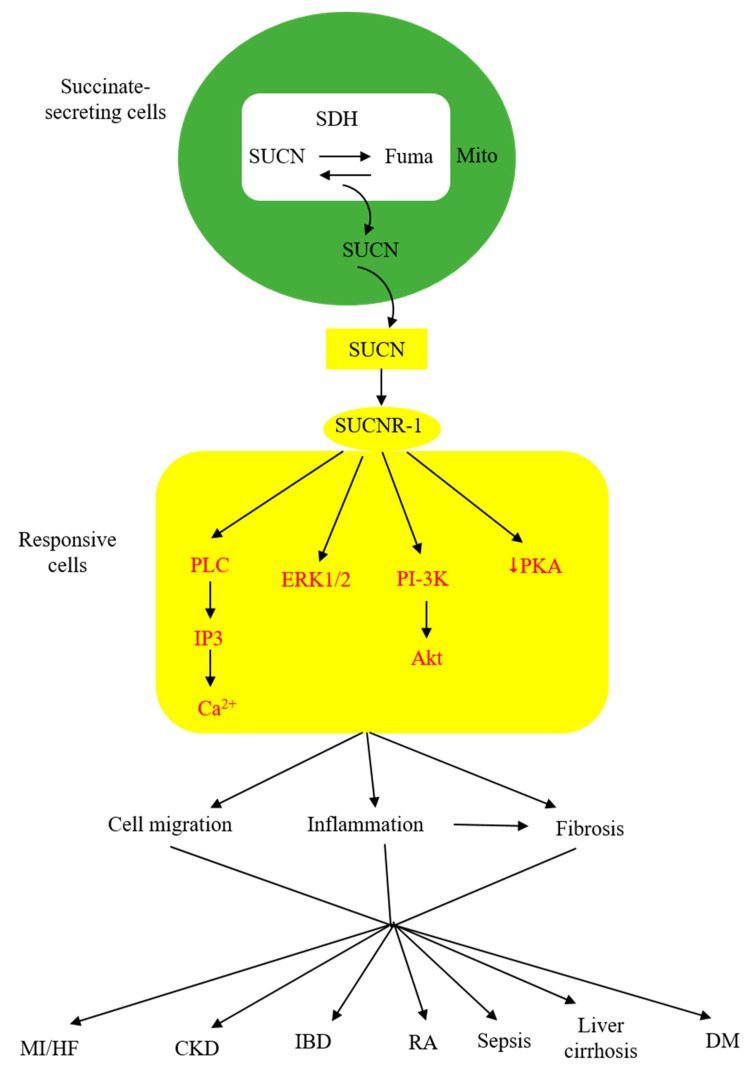
Schematic illustration of the involvement of succinate (SUCN)-SUCNR-1 axis in diverse pathological processes and human diseases. MI, myocardial infarction; HF, heart failure; CKD, chronic kidney disease; IBD, inflammatory bowel disease; DM, diabetes mellitus.

**Figure 3 ijms-24-11165-f003:**
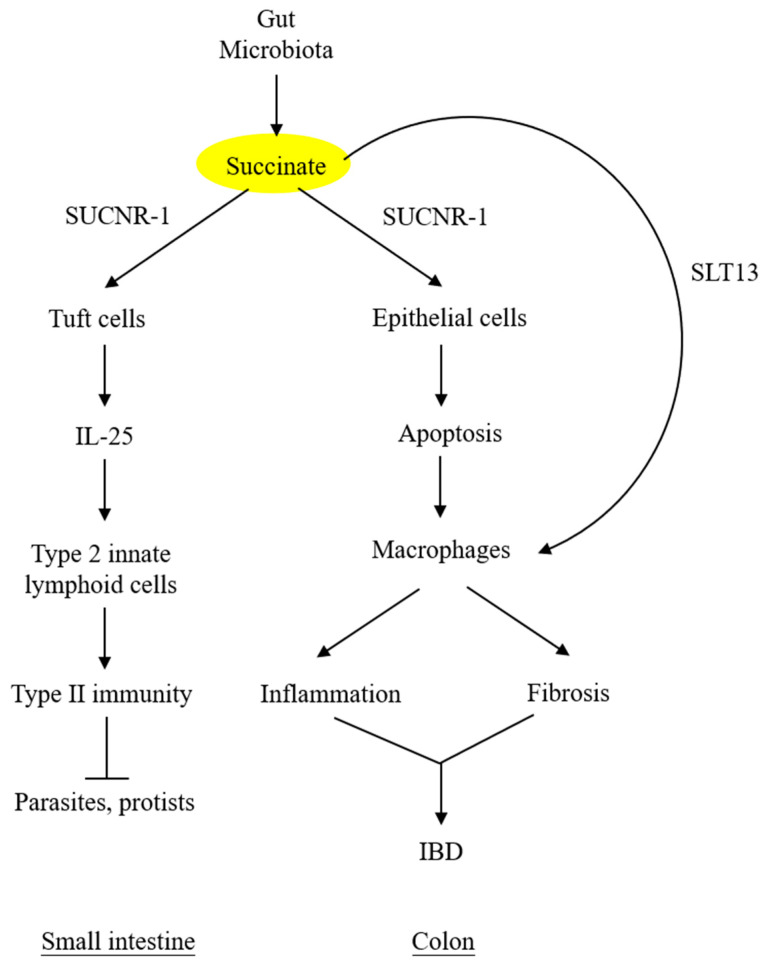
**Different roles of microbiota-derived succinate in small intestine vs. colon.** Succinate accumulation in colon exerts a direct detrimental effect on colon epithelium. It may also activate subepithelial macrophages via transepithelial transport. SLT13, solute transport 13; IBD, inflammatory bowel disease.

**Figure 4 ijms-24-11165-f004:**
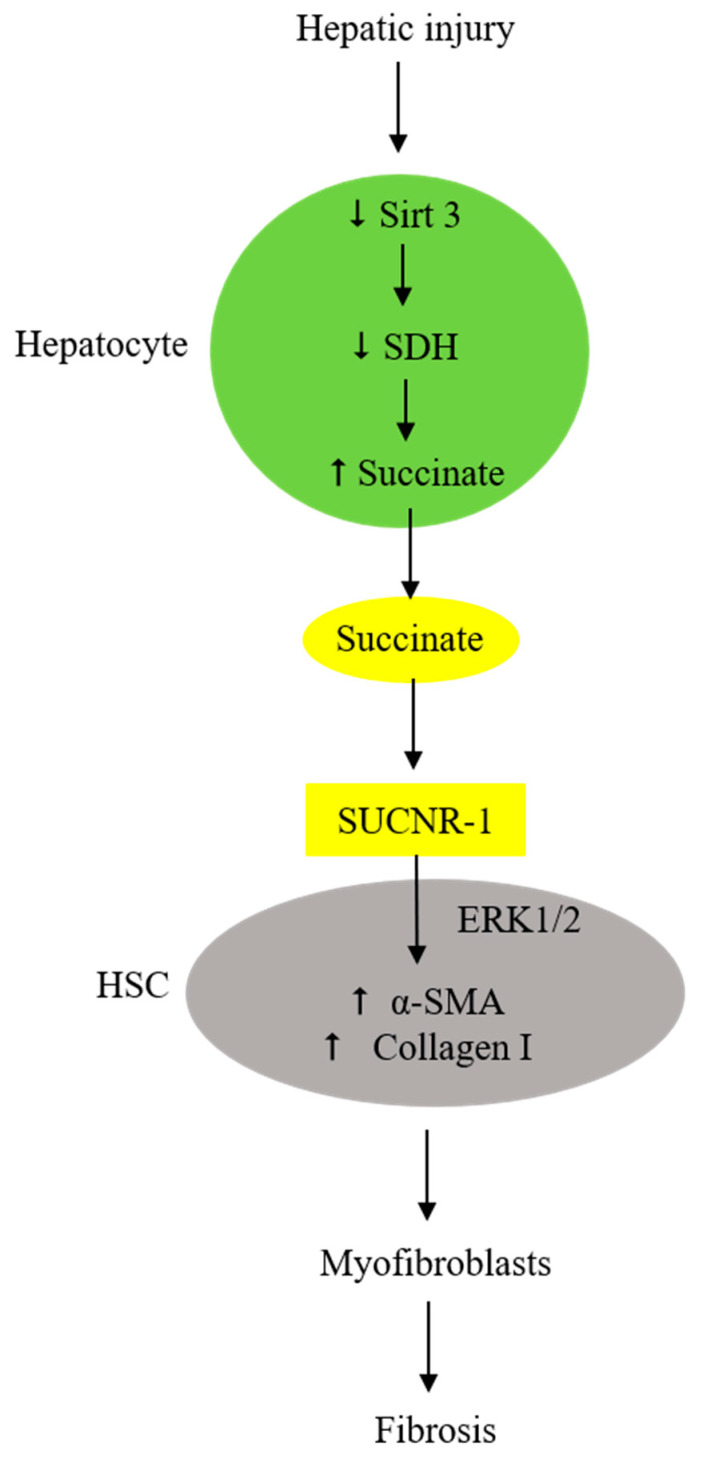
Extracellular succinate derived from injured hepatocytes induces HSC activation, myofibroblast differentiation, and fibrosis via SUCNR-1.

**Figure 5 ijms-24-11165-f005:**
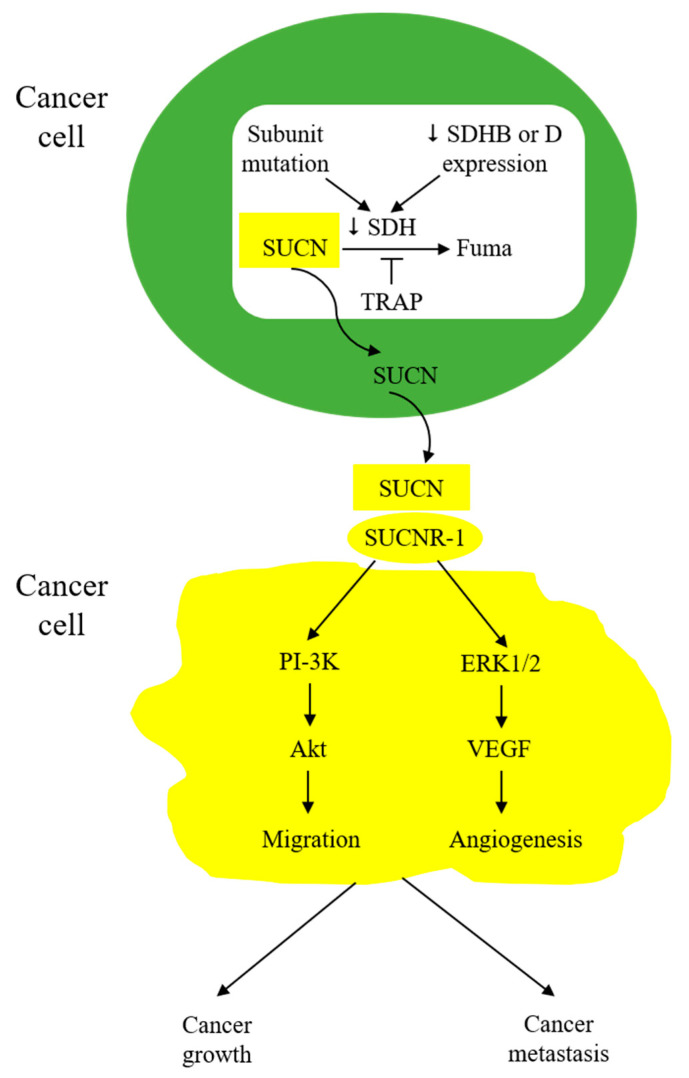
**Cancer-cell-derived succinate drives cancer cell migration and promotes cancer growth.** Succinate accumulation as a result of mutation of SDH subunits, notably B and D subunits, expression defects of SDHB or SDHD expression, or inhibition by TRAP. Succinate is secreted and acts in an auto- or paracrine manner to drive migration and growth via the signaling pathways, as indicated.

## Data Availability

Not applicable.

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
