# Peer review of "Extracellular Succinate: A Physiological Messenger and a Pathological Trigger"

_ijms, 2023, doi:10.3390/ijms241311165_

Round 1

Reviewer 1 Report

The author propose to summarize the current knowledge on extracellular succinate, focusing on multiple roles that this molecule plays as physiological messenger or as tissue killer, and trying to investigate the possible factors that can cause this switch. The review is well written, clear and exhaustive. It follows that reading is fluid and very interesting

Author Response

Appreciate the reviewer’s positive comments.

Reviewer 2 Report

The review entitled " Extracellular succinate: a physiological messenger and a pathological trigger " by Kenneth Wu summarizes the recent knowledge about signaling effect of extracellular succinate (EC-SUCN) in key organ. Circulating succinate can play two major roles on target cells that are a nutrient function with a cytosolic catabolism but also a signaling effect triggering SUCNR-1. The author well delimits the topic of the review focusing only on signaling effect in an introduction.

Then the review summarizes EC-SUCN/ SUCNR-1 axis in several chapter: skeletal-muscle physiological adaption with exercise and cold; small intestine and colon immunity; myocardial infarction; macrophages over-activation and organs diseases; hepatocyte-HSC dialogue and liver fibrosis; cancer progression.

Finally the review presents therapeutic strategies on SUCNR-1 and EC-SUCN availability to triggers the signaling effect of succinate.

The review looks exhaustive and interesting but need several form modifications.

1-Major change would be to totally focus on the EC-SUCN / SUCNR-1 axis  and not succinate substrate including in the title. Then remove in few writen parts when the author go on succinate entry into the cell, ie:

-       Remove 2.2 (l.74 to 102) as it deals with succinate substrate and its entry into BAT cells with SLC13 transporter.

-       Remove l.358 to l.361 and l.367 about intracellular succinate conclusion

-       Rewrite 9.2 and 9.3 that focus on EC-SUCN increase (by supplementation or SDH inhibition on cell transmitter respectivly) when targeting EC-SUCN/SUCNR-1 axis would be more to induce a decrease of EC-SUCN. Author can explain that targeting EC-SUCN by decreasing it would be detrimental for the other (ie substrate) role of succinate regarding the literature cited in 9.2 and 9.3

2-All figures highlight origin of EC-SUCN and target cells but need to be homogenized in color and form with the SUCN secreting compartment/tissue/cell and the triggered compartment (transmitter-receiver). A figure on cancer l.324 is lacking. Maybe one main figure regrouping all biological/pathological effect would be better. Thus, it would show the different signaling pathway activated by SUCNR-1 in the receiver.

3-The author should sometimes discuss autocrine/paracrine/endocrine effect of EC-SUCN when it is possible. Indeed circulating succinate coming from liver, muscle or other organ can act in other organ and sites of inflammation.

4- the review would be greatly enhanced by addition of concentration of circulating succinate and how it increases for each pathologies when known. To help on physiological: https://serummetabolome.ca or in serum vs organ in animal model doi: 10.1038/s41598-021-88097-8.

5-Other minor:

l.254: ref. 67 not 6 and 7

l.284-287: cite ref.

l.382: therapeutic agents “against” inflammation-related diseases

L.455, 456: use “Gi-protein” and “Gq-protein” mediated signaling pathway

L468-469: cite ref.

Author Response

Appreciate the reviewer’s comments. The reviewer raised several points about modifying the manuscript. Point-by-point response is given below.

  1. Per reviewer’s suggestion, the last paragraph of section 2.2 is deleted. However, the first paragraph of section 2.2 is retained as it gives this paper a comprehensive review of extracellular succinate (lines 74-87 of the revised manuscript). Statements in lines 358-367 of the original manuscript about intracellular succinate conclusion are deleted. In addition, sections 9.2 and 9.3 are deleted and section 9 is rewritten (lines 358-369).
  2. Per reviewer’s suggestion, figures are modified. Reviewer’s suggestion to include a figure “regrouping all biological/pathological effect” is well taken. Such a figure (Figure 2) is included in the revised manuscript (page 4). The original Figure 2 is deleted. The reviewer also suggested to include a figure on cancer. Such a figure is included in the revised manuscript as Figure 5 (page 11).

3 and 4. The comments are well taken. Appreciate reviewer’s suggestion to include succinate concentrations as well as references. A new section is added (Section 10 lines 386-410) and new refences are included (Ref #121-124; 126-127).

  1. Other minor. The errors are corrected.

Line 254 of the original manuscript: ref 6,7 is changed to 67 (line 246 of the revised manuscript).

Line 284-287: Two references are added, #73 and #74 (line 278 of the revised manuscript).

Line 382: “for” is changed to “against” (line 377).

Line 455, 456: Gi- and Gq- are changed Gi-protein and Gq-protein (line 430 of the revised manuscript).

Line 468-469: deleted.

Round 2

Reviewer 2 Report

The author makes all the suggested correction and I have no more comment. The new chapter 10 is nice.